# A Randomized Controlled Trial of a Parent Management Training Program for Incarcerated Parents: Post-Release Outcomes

**DOI:** 10.3390/ijerph19084605

**Published:** 2022-04-11

**Authors:** J. Mark Eddy, Charles R. Martinez, Bert O. Burraston, Danita Herrera, Rex M. Newton

**Affiliations:** 1Department of Educational Psychology and Department of Kinesiology and Health Education, College of Education, The University of Texas at Austin, Austin, TX 78712, USA; charlesm@austin.utexas.edu; 2Department of Criminology and Criminal Justice, University of Memphis, Memphis, TN 38152, USA; bbrrston@memphis.edu; 3Klamath Tribes Judiciary, The Klamath Tribes, Chiloquin, OR 97624, USA; danita.herrera@klamathtribalcourts.com; 4Department of Psychology, Portland Community College, Portland, OR 97224, USA; rexnewton@comcast.net

**Keywords:** incarcerated parents, families, prison, reentry, parent education, cognitive behavioral programs, randomized controlled trial, substance use problems, criminal behavior, recidivism

## Abstract

The majority of incarcerated adults are parents. While in prison, most parents maintain at least some contact with their families. A positive connection with family during imprisonment is hypothesized to improve long-term success after release. One way in which departments of corrections attempt to facilitate positive connections with family is through prison-based parenting programs. One such program, developed in collaboration with the Oregon Department of Corrections, is the cognitive-behavioral parent management training program Parenting Inside Out (PIO). Outcomes due to PIO were examined within the context of a randomized controlled trial. Incarcerated parents from all correctional facilities in the state of Oregon were recruited to participate, and eligible parents who consented (*N* = 359) were transferred to participating releasing institutions. After initial assessment, parents were randomized to condition (i.e., PIO “intervention” condition or services-as-usual “control” condition) and then followed through the remainder of their prison sentences and to one year after release. Intervention condition participants were offered PIO prior to their release. Outcomes favoring participants in the intervention condition were found in areas of importance to parents and their children and families and to public health and safety at large, including a decreased likelihood of problems related to substance use and of engaging in criminal behavior during the first six months following release as well as a decreased likelihood of being arrested by police during the first year following release. The implications of the findings are discussed, including the critical need for scientifically rigorous research on multi-component parenting programs delivered during the reentry period.

## 1. Introduction

Surveys by the federal government over the past quarter century have established that between 50 and 60% of incarcerated men and women in the USA are parents to young children [1,2,3]. According to the most recent such survey, the typical incarcerated parent has two minor-aged children, and the average age of these children is around 10 years old [3]. Many incarcerated parents (47% of fathers and 64% of mothers) report living with their children in the month prior to their incarceration. Most (75%) report that they were employed before their arrest, and many were the primary financial supporter of their families (54% of fathers and 52% of mothers). Clearly, many adults who enter prison not only have families, but they are connected to their families in vital ways. Given this, it is not surprising that continued connection to family during imprisonment is related to success after release [4,5,6]. Despite the numerous barriers families face in maintaining a relationship with a loved one in prison, almost 80% of incarcerated parents report having had at least some contact with their children since admission, including 39% of fathers and 56% of mothers who report at least weekly contact [2]. Evidence is accruing that parent–child contact during a prison sentence may be vital not only in terms of fostering a stronger connection between parents and their families after release, but also in terms of decreasing the likelihood of recidivism [7,8,9,10].

Departments of corrections have long attempted to deliver parent education programs [11]. A key target is fostering the quality and frequency of positive connections between incarcerated parents and their children and families during imprisonment. Such experiences are thought to lead to improved parent well-being and to set the stage for family social engagement and other forms of support following release. In turn, progress in these areas is thought to increase the likelihood of post-release success in domains important to individual, family, and societal health and well-being, including a reduced likelihood of criminal behavior and related problems, such as those related to substance use and abuse. 

While documentation is slim on the prevalence of prison-based parenting programs and the numbers of parents who receive them, it appears that most incarcerated parents are not offered the opportunity to participate [12]. However, in recent decades, awareness about the potential importance of these programs has grown, and efforts by multiple federal agencies have been made to document and disseminate information about promising policies and practices related to family contact in the hopes that more opportunities for connection would be made available for incarcerated parents [13]. During this same period, an increasing number of studies have been conducted that address incarcerated parents and their children and families [14,15,16]. 

One area of keen interest concerns the impacts of prison-based parenting programs on incarcerated parents and their children and families. The most recent comprehensive review of outcomes related to such programs was a meta-analysis by Armstrong and colleagues [17]. The authors identified 12,855 possible studies, of which 385 were found to be potentially eligible for the review. Ultimately, 22 studies were confirmed as being eligible, but only 16 studies (including four randomized controlled trials [RCTs]) provided sufficient information in their published reports to be included in the analyses. Across these, the common focus was on three outcomes immediately following intervention: parenting knowledge, parent–child relationship quality, and parent well-being. On average, small- to modest-sized effects were found on knowledge and relationship quality but negligible effects were found on well-being. 

One of the studies included in this analysis was conducted by Eddy, Martinez, Burraston and colleagues [18]. This RCT, known as The Parent Child Study, was a collaboration between researchers, the Oregon Department of Corrections (DOC), and the non-profit The Pathfinder Network and examined outcomes for 359 incarcerated fathers and mothers who were randomly assigned either to the intervention condition, Parenting Inside Out (PIO), a cognitive-behavioral parenting intervention explicitly designed for delivery to incarcerated fathers and mothers, or to a services-as-usual control condition. Immediately following intervention, while participants were still in prison, impacts favoring participants assigned to PIO were found on parenting, parent–caregiver relationships, and parent adjustment. Each of these constructs was a component of the Social Learning Theory model that undergirded the intervention, and each was directly targeted by the intervention. Depending on the outcome, impacts were of two types: main effects or moderated effects. When observed, moderation was in the form of a significant interaction between baseline levels of a given variable and condition, with parents who were more in need of intervention more likely to be impacted by PIO. 

Here, we report on outcomes for participants in The Parent Child Study after their release from prison. We focus on indicators of two upstream constructs in our theoretical model—parent adjustment and parent criminality. Each of these were hypothesized to be directly related to the parent–caregiver relationship, which in turn was hypothesized to influence the parent–child relationship and parenting. Positive outcomes for parents in terms of improved adjustment and reduced criminality are important not only because of their potential impacts on family relationships and day-to-day social interactions, but also because of their relationship with recidivism, and in particular a return to prison. Recidivism not only leads to the loss of opportunities for parents to have contact in the community with their children and families, but also the loss of opportunities to accrue other types of social and financial capital that are needed for long-term prosocial success during adulthood. 

### Hypotheses

**H1.** 
*Parents assigned to the intervention condition will be less likely to report substance use problems during the six months following release from prison than parents assigned to the control condition.*


**H2.** 
*Parents assigned to the intervention condition will be less likely to report engaging in criminal behavior during the six months following release from prison than parents assigned to the control condition.*


**H3.** 
*Parents assigned to the intervention condition will be less likely to be arrested by the police during the one year following release from prison than parents assigned to the control condition.*


## 2. Method

### 2.1. Design

This study took place within four minimum and/or medium security level adult correctional facilities operated by the DOC. Each of these was designated as a “releasing” facility, where incarcerated individuals were transferred near the end of their sentences to prepare for release back into their communities. One facility incarcerated women and three incarcerated men. Eligible and consenting participants were randomized within cohorts to experimental condition—namely the PIO intervention condition or the “services-as-usual” control condition—within each facility prior to the launch of a new series of PIO sessions. Before randomization, participants were blocked by race/ethnicity to ensure balance across condition. Throughout the course of the study, participants were invited to be assessed at multiple time points. In this paper, we focus on data collected at three time points: (1) baseline, prior to randomization (i.e., in person interview); (2) 6 months after release from prison (i.e., in person interview); and (3) one year after release from prison (i.e., official records collection). The study was approved by both the federal Office of Human Research Protections and the non-profit Oregon Social Learning Center Institutional Review Board.

### 2.2. Sample

To be eligible for study participation, an incarcerated adult was required to have at least one child between the ages of 3 and 11 years old, to have the legal right to contact that child, to have had some role in parenting that child in the past and expecting to have some role in the future (e.g., living with the child, regular contact with the child), to have not committed either a crime against a child or any type of sex offense, to have contact information (i.e., phone number, address) for the caregiver of that child, and to have less than 9 months remaining before the end of their prison sentence. Study recruitment took place in all DOC facilities across the state. Transfers were requested for eligible potential participants who did not reside in one of the four study facilities. Almost all such requests were approved. Of the 1483 adults who expressed interest in the study and were screened, 453 were deemed eligible. To ensure a demographically diverse sample, women and racial/ethnic minority participants were actively recruited and were oversampled from the eligible pool. From the invited potential participants, 359 (79%) consented to participate. These individuals, if not already residing in study facilities, were transferred to such, and randomization occurred after the initial assessment. Demographic characteristics of participants were as follows. In terms of gender, 54% were women and 46% were men. In terms of race/ethnicity, 59% of participants were White, 14% African American, 11% multi-racial, 8% Native American, and 7% Latino. In terms of education, 60% of participants had less than a high school education, 13% had a high school diploma or GED, and the remainder had at least some post-high school training or education. In terms of substance abuse, most participants (89%) had a history of drug and/or alcohol abuse or addiction. In terms of other mental health issues, many participants reported histories of such (37%), including at some point receiving a diagnosis of a mental health problem or a learning disability (48%). In terms of family, on average, parents had 3 children. Most children were biological children, and the average designated “target” child for the study (i.e., who was the focus of parent interview questions) was 8 years old. Prior to incarceration, 35% of parents had lived with their child full-time, 9% part-time, 18% visited with their child at least once a week, 14% visited less than once a week, and the remainder had infrequent contact. Further information about the sample and study recruitment is available elsewhere [18,19,20].

### 2.3. Experimental Conditions

**Intervention**. Participants assigned to the intervention condition were offered the PIO program. PIO is a 36-session version of evidence-based cognitive-behavioral parent management training (PMT) adapted for delivery to incarcerated parents. It was influenced by three major sources: (1) the PMT programs developed and tested at the Oregon Social Learning Center by psychologists Gerald Patterson, John Reid, Patti Chamberlain, Marion Forgatch, Beverly Fagot, Tom Dishion, and Kate Kavanagh and colleagues between the 1960s and the 1990s; (2) the programs designed by instructors throughout Oregon and the U.S. who were teaching parenting education in prisons at the time PIO was developed; and (3) the lived experiences of incarcerated parents and the caregivers of their children [11,21]. The program was created by a multi-disciplinary team that included clinical and developmental psychologists, therapists, and educators, and was designed to be delivered both to incarcerated fathers and to incarcerated mothers and to be culturally respectful. Program content included traditional PMT topics such as communication skills, positive reinforcement and involvement (including play), monitoring, discipline, problem solving, as well as child development, child health and safety, and personal and family decision making. Program processes included brief presentations, video clips, role plays, large and small group discussions, and class projects conducted both inside and outside of the classroom. Each session lasted 2 and ½ hours, and sessions were held three times per week for 12 weeks. Participants were encouraged to communicate with the caregivers of their child(ren) about sessions. Upon request by participants, caregivers received materials via the mail about each session (e.g., handouts that participants received). Caregivers were encouraged to talk with participants about their experiences in the session, as well as to contact program staff if they had any questions or needed local referrals for services. Sessions were taught by “coaches” who were employees of The Pathfinder Network, an agency with extensive experience delivering cognitive-behavioral programs within the DOC. With input from the research team, coaches were selected, hired, trained and supervised by Network staff. Supervision included individual and group supervision sessions, as well as the observation of randomly selected classroom sessions by a master trainer. Observations were of particular importance to the program. Following each observed session, the trainer would meet with the coach, discuss the session, and, if necessary, make a plan with the coach on how to improve teaching behaviors to be more congruent with the PIO model of intervention delivery. Active support for PIO was cultivated and maintained throughout the study at each level of the DOC, from the director of the system at large to the superintendents, correctional and program staff members of each participating institution. In this regard, a variety of agreements were made between the research team and the DOC that increased the likelihood of intervention integrity, from increasing the hourly pay of coaches to assist in decreasing staff turnover and cultivating expertise on the coaching team, to the establishment of policies that limited transfers out of facilities that were delivering PIO to increase the chance that intervention participants would complete the program. Additional information about PIO is available elsewhere [11,18]. Beyond PIO, participants in the intervention condition were allowed to access all other psychosocial services (e.g., substance abuse treatment programs, mental health treatment programs, parenting programs) that they were eligible to receive within their institution.

**Control**. Participants assigned to the control group were not allowed to enroll in PIO, but as with participants in the intervention condition, were allowed to access all other psychosocial services that they were eligible to receive within their institution. At times, these services included parent education programs, but these tended to be quite different from PIO in a number of important ways [11]. Such programs were typically delivered by the person who created the program and who was serving as an independent volunteer instructor (i.e., a person who was not working as part of an intervention team and who did not have an intervention supervisor). Furthermore, these programs typically were not evidence-informed in either their process or their content, and they tended to serve only a small number of parents.

### 2.4. Measurement

Participants were assessed via in person interviews at multiple time points, but here we focus on data from two interviews: baseline (i.e., before randomization and before the start of intervention) and follow-up (i.e., six-months after release from prison). Participants were compensated for their time for participating in each assessment, including USD 30 for the first in-prison interview and USD 100 for the out-of-prison interview. Interview participation was 100% at the baseline interview and 83% at the six-month post-release interview. In addition to interviews, official criminal justice records were collected on all participants one year following release from prison.

### 2.5. Dependent Variables

**Post release arrests.** The total number of police arrests during the first year after the release from prison was extracted from Oregon state-wide police records for each participant. Data were collected via OJIN OnLine, a web-based paid subscription data access system for court case information from all 36 circuit courts in Oregon, as well as from tax and appellate courts. The system enables searching for civil, small claims, tax, domestic, and criminal (including misdemeanor and felony) cases.

**Post release criminal behavior.** Criminal behavior during the six months following release was indexed via a version of the Elliott Social Behavior Questionnaire, a widely used self-report questionnaire that was administered via interview [22]. Participants were asked how many times they had participated in 39 antisocial and criminal behaviors, with answers ranging from 0 to 999 times. Items addressed a variety of areas, including theft, violent behavior and illegal drug sales, and were summed to create a measure of participation in criminal behavior. Cronbach’s alpha for the scale was 0.81.

**Post release substance use problems.** Problems related to substance use during the six months following release were measured with eight questions. Five of the items were dichotomously scored as 0 if “no” and 1 if “yes” and included “had to use more drugs or alcohol to get the same effect”, “drove or use equipment while using drugs or alcohol”, “desire to use drugs or alcohol so strong you couldn’t keep from using”, “experienced mental or emotional problems from drug or alcohol use”, and “had a month or more using drugs or alcohol or getting over using”. Two items used a 7-point scale (0 = “never” to 6 = “daily”) and queried “how often substance use interfered job or home” and “how often used larger amounts of drugs or alcohol”. The final item was “how much has drug or alcohol use caused problems” and was measured on a 10-point scale, with 0 indicating “no problems” and 10 indicating “many problems.” The final score was the sum of the eight items. Cronbach’s alpha for the scale was 0.86.

### 2.6. Independent Variables

**Condition.** Randomly assigned intervention condition is the primary variable of interest. Condition is a dummy variable coded “1” for all subjects assigned to the PIO intervention condition and “0” for all subjects assigned to the services-as-usual control condition.

### 2.7. Control Variables

**Prior arrests.** The total number of police arrests prior to the current sentence of a participant was measured according to the Oregon police records collected via OJIN OnLine. Data were available for 345 participants only; 14 participants refused to consent to a search of their prior arrest records (i.e., 3 control participants, 11 intervention condition participants).

**Prior criminal behavior.** Criminal behaviors during the year before the current incarceration were measured during the post-intervention interview using the same items from the Elliott Social Behavior Questionnaire (see above) used to index criminal behavior after release. Cronbach’s alpha for the scale was 0.83. Since prior criminal behavior had a skewed distribution, prior to analyses, the distribution was normalized using a natural log transformation.

**Prior substance use problems.** Substance use problems during the three months prior to incarceration were measured during the post-intervention interview. Participants were queried with the same eight questions that were used to measure post release substance use problems as well as a ninth item of “have you ever had a serious problem with drugs or alcohol” (0 = “no”, 1 = “yes”). Cronbach’s alpha for the scale was 0.86.

**Years incarcerated.** The DOC provided a sum of the total number of years the parent had been incarcerated prior to the start of the study. Since years incarcerated had a skewed distribution, prior to analyses, the distribution was normalized using a natural log transformation.

**Gender**. Women were coded “1” and men were coded “0”.

**Mental health problems**. The DOC provided a count of the total number of mental health problems and learning disabilities noted in their records during each participant’s sentence prior to the start of the study. From this information, this variable is dummy coded “1” if the participant had been diagnosed with either a mental health problem or a learning disability and “0” if a participant had not received any such diagnosis.

**Family contact.** Family contact in prison was a self-report index of the total number of in person, phone, and letter contacts by family members during imprisonment in the month prior to the baseline assessment.

### 2.8. Analytic Strategy

We examined intervention effects on the dependent variables using STATA Zero-Inflated Negative Binomial Regression (ZINB), which allowed us to control both for the over inflation of zero in the count outcomes and for over dispersion [23,24]. In addition, we used adjusted standard errors corrected for the nesting of participants within correctional facilities. An intent-to-treat approach was used in each analysis: intervention condition participants were included in analyses regardless of their attendance and/or participation in the PIO sessions. For each outcome, we tested a ZINB regression model for differences by condition controlling for baseline levels of the outcome variable, gender, total time in prison, family contact, and mental health problems. Each of these factors is related to post-prison outcomes [14,16]. In predicting the zero-inflation portion of each model, we used condition, gender, and the baseline measure. In each portion of the model, nonlinear effects were included when significant. To address missing data, we used the multiple imputation procedure [25,26,27,28] available in STATA to impute both missing independent and dependent variables [29]. For each missing value, we imputed 50 values and then used the mean of these values as the final imputed value.

## 3. Results

### 3.1. Intervention Delivery

Most participants (94%) who were randomly assigned to the intervention began the program. Of these 182 parents, participants attended an average of 24 out of 36 sessions, with 66% attending at least 20 of sessions. In terms of program fidelity, on average, 90% of the content in the curriculum was taught in each session. In session observations, PIO coaches received an average of 3.9 out of 5 (with 1 “below expectations” to 5 “exceeds expectations”) across 32 questions on expected session processes (e.g., content delivery methods, discussion facilitation, support and encouragement of participants). Further information on intervention delivery during the RCT is available elsewhere [18].

### 3.2. Participant Satisfaction

Participants in the intervention condition viewed their experiences in the PIO program favorably. Most participants indicated that they would “strongly recommend” the program to other incarcerated parents (on a 5-point scale, average of 4.5, median of 5). Furthermore, 70% of parents rated the information they received as “very” or “quite” helpful, 90% of parents rated the program as having a positive effect on them, and 95% rated the program as being useful to them as parents.

### 3.3. Intervention Outcomes

Descriptive information on the variables in the analysis are provided in Table 1. The ZINB regression coefficients, incident rate ratios (IRR), and significance levels for the predictors in the models that were examined are listed in Table 2. Of note, when employing ZINB regression, there are two ways to detect an intervention condition effect. The first is for the intervention to have a significantly lower incident rate ratio (e.g., participants in the PIO intervention condition engage in fewer criminal activities). The second is for the intervention to significantly increase the likelihood of a participant abstaining from the activity of interest (e.g., no self-reported criminal behaviors). Therefore, in the three models, what was of most interest was whether intervention condition significantly influenced each incident rate ratio and the probability of the outcome being zero. For all three outcomes (i.e., post-release arrests, post-release criminal behavior, and post-release substance use problems), we found significant main effects of the intervention. Namely, intervention condition participants had significantly fewer arrests and were more likely to abstain from criminal behavior and problematic substance use. Findings by outcome are described below. For each outcome, findings were similar whether or not an imputed dataset was used in the analysis.

**Post-release arrests.** There was a significant main effect of condition on arrests through one year after release from prison (see Table 2). The model explains about 20 percent of the variation in post-release arrests. Controlling for prior arrests, gender, and time in prison, participants assigned to the intervention condition had 37 percent (IRR = 0.63, *p* < 0.05) fewer arrests than control participants through one year after release from prison. Women had 48% (IRR = 0.52, *p* < 0.001) fewer arrests than men. For every unit increase in prior arrests, the IRR increased 10.03 times (*p* < 0.01). For each year increase in time in prison before this incarceration, the IRR increased 1.58 times (IRR = 1.58, *p* < 0.001). For each unit increase in family contact, the IRR decreased on average by 2 percent (IRR = 0.98, *p* < 0.06). Those with mental health problems had 77 percent fewer arrests than those who did not (IRR = 0.23, *p* < 0.001). Further, total prior arrests and gender both significantly predicted the zero inflation. Women were significantly more like to have no arrests compared to men (*b* = 0.17; *p* < 0.001).

**Post-release criminal behavior.** There was a significant effect of condition on criminal behavior to six months post-release (see Table 2). The model explains about 13 percent of the variation in post-release self-report of criminal behavior. Intervention, prior criminal behavior, and gender all significantly predicted the zero inflation. Controlling for prior criminal behavior and gender, intervention condition participants were significantly more like to report no post-release criminal behaviors (*b* = 0.40; *p* < 0.05). Women were more likely to report zero behaviors than men (*b* = 2.10; *p* < 0.001). The more prior criminal behavior, the less likely a person abstained from post-release criminal behavior (*b* = −0.41; *p* < 0.001). Furthermore, controlling for prior criminal behavior, gender, and years incarcerated, participants in the intervention condition reported 30 percent (IRR = 0.70, *p* > 0.05) fewer criminal behaviors than control condition participants through six-months after release. This finding, however, was not significant. Prior criminal behavior was a significant predictor at the cubic level, a finding discussed elsewhere [30]. For each family contact, post-release self-report of criminal behavior decreased on average by 3 percent (IRR = 0.97, *p* < 0.05). Those with a mental health problem reported 61 percent fewer criminal behaviors than those who did not have a mental health problem (IRR = 0.39, *p* < 0.05). Gender and time in prison did not have significant incident rate ratios.

**Post-release substance use problems.** There was a significant effect of condition on substance use problems through six-months post release (see Table 2). The model explains about 8 percent of the variation in post-release self-report of substance abuse problems. Condition and gender both significantly predicted the zero inflation. Controlling for prior substance use problems and gender, intervention condition participants were significantly more likely to report no post-release substance abuse problems (*b* = 0.44; *p* < 0.001). Women were more likely to report zero problems than were men (*b* = 1.29; *p* < 0.05). Controlling for condition and gender, prior substance use problems were not significantly related to the zero inflation. Further, controlling for prior substance abuse problems, gender, and years incarcerated, participants assigned to the intervention condition had no differences in substance use problems IRR compared to the control group (IRR = 10.07, *p* > 0.05). Post-release substance use problems increased 10.05 times for every unit increase in prior substance use problems (*p* < 0.05). Gender, time in prison, family contact, and mental health problems were not significant predictors.

## 4. Discussion

Parents assigned to the intervention condition were significantly less likely to report problems related to substance use and engaging in criminal behavior during the first six months following their release from prison. Given this, it is not surprising that based on official state records, parents in the intervention condition also were significantly less likely to be arrested during the year following their release from prison. These positive findings built on earlier reported findings [18], including that most parents assigned to the intervention condition participated in most PIO sessions, that measures of fidelity indicated that the intervention they received was delivered as planned, that most parents felt the program was useful to them as parents and had a positive effect on them, and that positive impacts of PIO were observed on parenting, parent–caregiver relationships, and parent adjustment while parents were still incarcerated.

While these findings are promising, and they extend the foundation of knowledge that has been built regarding prison-based parenting programs over the last few decades, this foundation is unstable. As is clear in the Armstrong et al. meta-analysis [17], quality research evidence is scant on prison-based parenting programs. To shore up the infrastructure underlying the field, scientifically rigorous studies of prison-based parenting programs must be conducted that include not only sound designs, such as the RCT, but also quality improvements in all areas of those trials, including but not limited to sampling, sample size, measurement, intervention fidelity, and extended follow-up after release from prison. RCTs that attempt to extend promising findings regarding parenting programs that are currently being used in prison settings are particularly needed.

Within such studies, a topic that is particularly in need of illumination is an answer to the question that the field of intervention research at large seems reluctant to address: “What treatment, by whom, is most effective for this individual with that specific problem, and under which set of circumstances?” [31]. Instead, the question of interest is typically “Does it work?” As noted by Paul [31], this question is probably the least important question that can be asked in intervention research. As practitioners find out quickly when they try to deliver “evidence-based” programs in the field, having an answer to this popular question, and this one alone, means one knows relatively little about what to do when your version of delivery appears not “to work” with a particular population in a particular setting. Answering the more complicated question of Paul [31] requires that researchers not only examine outcomes due to interventions within more rigorous designs, but also that their studies are sufficiently powered to detect the impacts of moderating variables on key outcomes. This requires not only larger samples, but higher-quality measurement and longer follow-up periods than are typical in the field thus far.

Finally, information is needed on the role that prison-based parenting programs might play in the context of a comprehensive approach to the reentry of parents back into their communities and their families. As we have discussed elsewhere [32], the role of a parent is more complex than having a certain set of parenting knowledge and skills and being able to communicate well with the various adults in the life of a child. These are necessary, but not sufficient, to support a child on their journey from infancy to adulthood. In our current work, we strive to focus on two sets of primary targets in reentry programs for parents. The first set, the “Little Four”, are vital to a well-functioning parent–child relationship: positive involvement, supervision and monitoring, guidance and discipline, and problem-solving. The second set, the “Big Four”, are vital to the well-being of the parent and their family: parent mental and physical health, safe and stable housing, a living wage job, and the quality of parent-other adult relationships. We posit that success in each of these areas is needed for the future success of formerly incarcerated adults and their children and families. Because the Big Four are so important, what is of most interest to us is whether or not parenting programs have added value over and above programs that are successful in promoting the Big Four, and if so, for whom. Such programs are most likely to succeed, we hypothesize, if they start in prison and continue after release [33,34,35]. This is not to suggest that programs that target only the Little Four cannot also have important impacts, but rather that if major failures occur in the Big Four, families, and thus children, are at high risk for problematic outcomes.

As with all studies, this study has a variety of weaknesses. Although this is, by far, the largest RCT of a prison-based parenting program, even larger samples with greater diversity are needed to examine the plethora of moderation questions that need to be examined. While we set out originally to measure a variety of constructs related to participant and family functioning and from multiple points of view, it turned out to be quite difficult to recruit caregivers and even more difficult to recruit children of incarcerated parents [18]. Better representation of the points of view from each of these groups would have improved the quality of measurement. In terms of the intervention, it was designed to meet parameters that were requested by our corrections partners, and PIO ended up being rather lengthy compared to typical parenting programs. This made it more difficult to deliver the entirety of the intervention to participants. Just as for parents in the community, there are many reasons that incarcerated parents are not able to attend sessions, including work duties, health status, and scheduling conflicts, and thus the typical parent only attended around 70% of the sessions. Ideally, parents would receive the full “dose” of the intervention. Since this study, and in response to requests by other corrections departments, we have created briefer versions of PIO, including for delivery in jail and community (e.g., post-prison supervision, probation, parole) settings, but outcomes due to these have yet to be examined in a RCT.

In conclusion, the findings reported here, in combination with the findings in the previous report [18], indicate that an evidence-informed parent management training program designed specifically for delivery to incarcerated parents has promise in terms of generating positive impacts of importance to parents and their families. In making such a statement, it is important to keep in mind what the word “program” really means in practice. A program is a living, multi-layered entity. It comprises not only the process and the content of the program—the curriculum—but also how it is delivered. In the case of PIO, delivery is accomplished through the ongoing actions and interactions of parent coaches who are carefully selected, trained, supervised, and supported. Yet “delivery” is much more complicated than this. Coaches work within the correctional environment, and continual support of coaches from influential people within that environment is vital for success. Support begins at the level of the headquarters, but without even stronger support at all levels within a correctional facility, from the superintendent to correctional officers to other prison staff members, the goals of a parenting program cannot be achieved. In turn, support from these individuals to the parents who are participating in a parent program is key to success. For example, we have heard program participants comment time and again that some of the first social interactions in which they try out the evidence-informed speaking and listening skills that are central to PIO are with correctional officers. If parents experience repeated success in their moment-to-moment interactions with prison staff, the impact can be powerful and set the stage for future positive interactions with caregivers and children. In short, the success of a prison-based parenting program is a community affair, and one that has the potential to initiate a chain of positive events that can have a profound impact in the life of a parent, their child(ren) and family, and their community. Our hope is that the promise seen in this study will generate excitement for infusing research-based knowledge into prison-based parenting programs and for conducting additional RCTs of research-informed programs. Such advances are needed to create a deep evidence base of knowledge that will enable parenting practitioners to optimally serve the wide variety of families impacted by incarceration. The ultimate beneficiaries of such progress are children and society at large.

## Figures and Tables

**Table 1 ijerph-19-04605-t001:** Descriptive statistics.

	Intervention		Control	
Variable	*M* (Count)	*SD*	*M* (Count)	*SD*
*Dependent Variables*				
Post-Release Arrests	1.35	3.77	1.41	4.01
Post-Release Criminal Behavior	8.97	25.61	12.37	29.23
Post-Release Substance Abuse	2.91	6.08	3.60	6.37
*Independent Variable*				
Intervention Condition	(194)		(165)	
*Control Variables*				
Prior Arrests	17.21	13.44	16.61	12.37
Prior Criminal Behavior Logged	4.70	2.35	50.02	2.23
Prior Substance Problems	13.21	9.33	14.91	8.68
Years Incarcerated Logged	−0.06	0.90	−0.09	0.87
Family Contact	4.67	7.55	50.01	8.34
Women	(107)		(91)	
Mental Health Problems	(12)		(9)	

**Table 2 ijerph-19-04605-t002:** Zero-inflated negative binomial regression models.

	Total Arrests Through One Year Post-Release	Criminal Behavior Six Months Post-Release	Substance Problems Six Months Post-Release
	Coefficient (IRR)	Coefficient (IRR)	Coefficient (IRR)
Count Condition ^a^	−0.47 * (0.63)	−0.36 (0.70)	0.07 (1.07)
Baseline ^b^	0.03 ** (1.03)	−1.60 *** (0.20)	0.05 * (1.05)
Baseline Squared Baseline Cubed	--- ---	0.42 *** (1.53) −0.03 *** (0.97)	--- ---
Women ^c^	−0.66 *** (0.52)	−0.15 (0.86)	−0.05 (0.96)
Years Incarcerated (Logged)	0.46 *** (1.58)	0.26 (1.30)	0.05 (1.05)
Family Contact (Logged)	−0.02 (0.98)	−0.03 * (0.97)	0.00 (1.00)
Mental Health Problems	−1.46 *** (0.23)	−0.93 * (0.39)	−0.13 (0.88)
Constant	1.03 ***	3.45 **	0.96 **
Inflation			
Condition	−0.25	0.40 *	0.44 ***
Baseline	−0.07 ***	−0.41 ***	0.02
Baseline Squared	0.005 ***	---	---
Women	0.17 *	2.10 ***	1.29 *
Constant	1.39	−0.46	−1.05
Ln(alpha)	−0.26	1.69 ***	0.68 *
Alpha	0.77	5.45	1.97
Maximum Likelihood R^2^	0.20	0.13	0.08

Notes. * *p* < 0.05. ** *p* < 0.01. *** *p* < 0.001. ^a^ = control condition is the reference group. ^b^ = baseline year prior to incarceration total number of criminal behaviors is natural log transformed. ^c^ = men are the reference group.

## Data Availability

The data presented in this study are part of ongoing work by the research team. Data are available on request from the corresponding author.

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
