# Peer review of "A Randomized Controlled Trial of a Parent Management Training Program for Incarcerated Parents: Post-Release Outcomes"

_ijerph, 2022, doi:10.3390/ijerph19084605_

Round 1

Reviewer 1 Report

Dear authors, I would like to congratulate you for your work, which is of great theoretical and applied interest.
First of all, I would like to point out your merits. And finally, some suggestions that could improve the final presentation and that are easy to consider. 

This paper presents the long-term outcomes of the cognitive-behavioral parent management training program “Parenting Inside Out” (PIO) within the context of a randomized controlled trial (RCT). Participants were 359 incarcerated parents from all correctional facilities in the state of Oregon. The results obtained in the follow-up phase (six months post release) are presented. The effects are compared with those obtained with a control and an alternative intervention program. The authors conclude that the outcomes favoring participants in the PIO condition in a variety of areas, including decreased likelihood of reporting substance use problems and engaging in criminal behavior during the first six months following release and decreased likelihood of being arrested by police across the first year following release than the control program. The program presents the theoretical starting model and references the publications in which the design of the program in question can be found.

There are two applied-theoretical issues that bring relevance to the present study. The first is the emphasis on developing an intervention program based on theory and evidence of outcomes. Second, the interest on the evaluation of intervention programs, while not novel in themselves, remain a minority in social intervention. Therefore, I consider that this work constitutes a contribution to the knowledge of the field of study.

From an applied point of view, it proposes an intervention program with solid theoretical and empirical support and whose design, planning and execution reflect great care and respect for all those involved. The hypotheses, analysis and discussion are solidly grounded and the presentation of the results and discussion are appropriately adjusted to the design of the work.

On the other hand, the work is written appropriately, clearly, and concisely and focused on the fundamental aspects that clarify the objectives of the study and its relevance.

It is a work of great scientific soundness. The methods and tools of analysis are appropriate for the objectives formulated and provide adequate evidence to analyze the study hypotheses. The analyses performed make the long-term effects of the program visible in a simple and clear way. The authors also appropriately point out the limitations of the work. They do so in a straightforward manner and without minimizing their importance. 

I believe it is a work that will arouse great interest from different disciplines, and both among theoretical and applied scientists .

I consider that the work can be published, although there are some minor issues that should be corrected.

- Line 94. Missing Figure 1

- References not included:

- Elliot 1983 cited in line 227 is cited.

- Elliot, 1994,cited in Lines 255-256

- Lines 227 et seq. and lines 254 et seq. There are discrepancies in the description of the instruments for measuring PRE and POST criminal behavior. Justify: why they are different instruments.

- Lines 226 and 254. Recall on these lines (and on the rest of the instruments if appropriate) the time at which this record was taken - six months after release from prison. It is noted in the Abstract, but It's not superfluous to recall it in this section.

- Lines 226 and 255 in the measurement instrument reference, there is information that appears to be discrepant. I suggest completing the description of the PRE instrument by clarifying that the instrument used, Elliott Behavior Checklist (Elliot, 1994), was a version of the Elliott Delinquency Scale (Elliott, 1983).

Author Response

  1. The reference to Figure 1 in the introduction and background section of the paper was in error. There is no Figure 1. We deleted this text.

  1. We apologize for the confusion regarding the Elliot reference. We have corrected it in the text and the reference section.

  1. This refers to the Elliot reference noted in (2). We corrected this.

  1. We added text about the time frame for the outcome variables (i.e., see lines 127-131; lines 210-219; and for each variable noted in lines 210-278).

  1. This refers again to the again to the reference noted in (2). We corrected this.

Reviewer 2 Report

Other than to express positive and high regard for the attention to detail and methods within which this study was carried out and the manuscript drafted, I do not have much to offer in terms of comments and suggestions.  I believe the manuscript represents a comprehensive representation of the study, including the introduction, establishment of relevance and context, review of the literature, methods, findings and discussion.  It is important for me to share that I possibly should not have accepted this review task, because I am not a quantitative researcher and have no real expertise in quantitative research method and design.  That said, I was unable to provide and evaluation of and/or comment on the study's methods.  What I can share is that as someone without an extensive background in quantitative research, I was still able to follow the explanations provided and develop some understanding of the research design.  I was also able to understand the findings and conclusions drawn.  I share this to say that the paper is well written for someone at the beginner stage of research analysis and review, and as a reviewer I appreciated that.  In terms of written quality, the external and internal structure of the paper is well organized, word choice and phraseology are also well done.  Tables and figures compliment the text, and though I am unable to offer any meaningful commentary on the descriptive statistics, figure presentations complimented by the narrative made it easy to follow and understand.  The most powerful part of the paper was the discussion section.  The authors do an extraordinary job of explaining findings.  Any questions I had about sample size, participant demographics, and/or societal factors that may prove relevant to findings were addressed here.  I was particularly grateful for the discussion of "little four" and "big four" and what these concepts mean for both this study and future research.  Overall, this study and  manuscript was well done.  I am not versed enough in the body of work surrounding parent programs for incarcerated individuals to offer insight into the literature reviewed, however, the reference section appears to have a balance of studies across a relevant time span (mid 1900's through 2021).  Overall, well done.

This may seem like an attempt to circumvent providing a comprehensive review but please understand that it is not.  This manuscript was one of the best reviews I have undertaken in some time.  It was easy to follow, the topic is relevant, the work is well written with few if any grammatical or spelling errors, and though this is not my core field, I was genuinely interested in the content from beginning to end and found myself reflecting on the piece in terms of its connection to my work and interest in the the school-to-prison pipeline.  I understood this study to rely primarily on self-reporting in terms of the parent-child-family relationship and I wondered what a study of this sort might look like centered in youth voice.  Thanks for sharing.  Well done.

Author Response

  1. We noted in the submitted draft the importance of including reports from a variety of agents, including children. Due to this comment, we added some language to further emphasize the utmost importance of giving voice to different points of view.

Reviewer 3 Report

Overview and General Comments

    Strengths:

1. This manuscript presents the results of a 6 month outcome study of a prison based cognitive behavioral parenting program, Parenting Inside Out (PIO) to improve long term success after release.

2.The method was a randomized controlled trial and appropriate to the study design.

   Weakness:

The overarching question is whether or not this study substantively adds to the field, specifically in light of similar studies published by this research team in peer reviewed journals and book chapters (available and in press).  For instance, the one year outcome study was published nearly 10 years ago (2013) and only a general study comparison was presented in the first paragraph of the discussion. Please clarify what this particular study uniquely adds to the subject of interest compared to your other manuscripts. This may be further clarified in the introduction as well as the discussion. Specifically what distinguishes the 6 month and one year outcome studies with respect to the primary outcomes.

Introduction/Hypotheses

  Strengths:

1. Well written with appropriate references.

2. The background literature review generally supported the purpose of the study and the hypotheses.

 Weakness:

Line 77. “Unfortunately, because of the focus on these reports on outcome alone.” This statement is unclear. Perhaps it was supposed to be the first part of the sentence that follows on line 78.  Please revise

Method

1. The study design is optimal for intervention outcome research.

2. The intervention (PIO) is well developed and evidence-based; implemented by “coaches” with good training, experience and supervision; and appropriate for the study aims.

    Weaknesses:

  1. Line 124: Please provide more information on the “services as usual control condition” such as what type of services were provided and to whom.
  2. Lines 127-128: “…participants were invited to be assessed at multiple time points before and after release from prison” Please consider substituting “specified time points” or “predetermined multiple time points” for multiple points.
  3. line 220: You collected post release arrest data over the first year, but your outcome measure was previously described as assessed at 6 months. Please clarify that this was part of a larger outcome study.

  Results:

  Strength:

     Straightforward and well described using appropriate analytical               methods.

  Weakness:

   Page 8, Table 2: The columns seem to be out of alignment. Please correct.

  Discussion

Strengths

  1. Overall Interpretations and conclusions were reasonable in light of the results.
  2. Study limitations or weaknesses were adequately addressed.

  Weakness:

     As noted above, please provide a comparison between the six and one month outcomes on primary outcome measures.

Author Response

  1. The Eddy et al. paper published in 2013 focused solely on data collected before participants were released from prison but after intervention was completed. In contrast, the current manuscript reports on outcomes after participants were released from prison, and provides all the data available from this study on two very important outcomes to incarcerated parents – substance abuse and criminal behavior.

This current manuscript differs from the other papers that have been published. The Eddy et al. (2008) paper described the development of the Parenting Inside Out (PIO) program. The Kjellstrand et al. (2012) paper and the Borja et al. (2015) present information on demographics and historical experiences (e.g., trauma) for participants as measured at baseline, before randomization. The various chapters concern broad issues related to parenting programs, and particularly discuss the notion of multimodal intervention programs for parents, of which PIO is not.

This current manuscript is the first to present outcomes indexed at six months after release from prison. However, if published, it would be the second paper to present findings related to one-year post-release data, but the earlier paper (Burraston & Eddy, 2017) was written to illustrate a certain type of analysis within a special issue on incarcerated parents, rather than to present intervention main effects (i.e., the purpose of the present submission).

Due to my research with incarcerated populations and their families over the past twenty-five years, I am quite familiar with the intervention research with incarcerated parents, and this manuscript does make a substantive contribution. The current study is unique in this genre of work, not only in the size of the sample but also in the outcomes examined and in both the length and settings (i.e., in prison, out of prison) of follow-up. This area of research is very much in need of an upgrade in quality and scientific rigor, and this study represents progress toward where I hope the science in this area can go next – from moving beyond pre-post and quasi-experimental evaluations with a quite limited set of outcomes to conducting randomized controlled trials that include multiple follow-ups after release from prison with a broader set of outcomes reported on by multiple agents.

  1. The confusing sentence was deleted around lines 77-79. The point that was being made there is not important to the content of this paper.

  1. Additional information was added between lines 206 to 215 regarding the services as usual control condition. Unfortunately, information about the other psychosocial services received by participants in either in the intervention or the control condition is not available. These were free to vary across all participants.

  1. A clarification to the language regarding assessments were provided around lines 127-128.

  1. A clarification was made regarding the 6-month and one-year confusion that was noted on line 220.

  1. I had difficulty in the document that was returned to me in correctly aligning the various columns in the table. Work is needed in this regard. If I should hire an editor to assist on this, please let me know as soon as possible.

  1. We note the time frames for the various outcomes in the discussion section.

Reviewer 4 Report

Itroductin

There was no figure 1 in the article or in the additional materials.

Could the authors indicate the main reasons for the failures of rehabilitation programs?

  1. Method

How operationalized: „to have had some role in parenting their children in the past”? line 135. Pease explain it

Did the inmates while serving their sentence participated in programs counteracting drug addiction? Was this variable controlled?

2.5. Dependent Variables

Is it possible that inmates who left the Correctional Facility would later commit a crime and it was not registered in the databases used by the authors? If so, please include this information in the discussion.

  1. Results

To what extent, in the authors' opinion, the self-report measures accurately represent the actual illegal behavior of former inmates?

  1. Discussion

What psychological variables can characterize the people who participated in this program?

How can the impact of such programs on inmates be improved?

Author Response

  1. We deleted the reference to Figure 1. There is no such figure.

  1. This paper is not focused on reentry programs at large. Further, there is not a body of literature that exists on how well parenting programs conducted in prison per se impact long term outcomes. Thus, there are not data that can be garnered to say whether or not rehabilitation programs that include a focus on parenting are likely to be a failure or a success, let alone why.

  1. We added additional text related to the role that a parent played with their child prior to incarceration (see around line 135). This study was intended to focus on a sample of parents who had a least some connection to their children and knew who their child lived with and how to contact the caregiver at the time of recruitment. What the specific type and frequency of contact was like was left free to vary.

  1. The only thing that was randomized in this study was participation in the PIO program. Other services were available to all parents regardless of whether or not a participant was assigned to the intervention or to the control condition. Substance abuse treatment programs were available in the prisons where the study was conducted, and some mothers and fathers in each group participated in such programs. Certainly, a history of substance use, and often of substance abuse, is common in incarcerated populations in the U.S. Unfortunately, we did not ask about participation in substance abuse treatment during the current incarceration, nor did we request Department of Corrections records about such, and thus we do not have a variable in this data set that documents recent treatment participation. Of note, this issue can get complicated and confusing quite quickly, given the high prevalence of substance use problems in this sample (over 85%; see Kjellstrand et al., 2012) and that some participants had served multiple prison sentences. Some mothers and fathers may have participated in these types of treatment programs multiple times in prison, let alone after release. Hopefully, randomization worked in terms of equal numbers of participants across the two conditions. Further, we did control for total time incarcerated in the analyses.

  1. Yes, it is possible that crimes were committed by research participants that were not logged in the state database. This could be due to a number of reasons, including a person not being arrested by the police in relation to the suspected crime, or the arrest of a person outside of the state of Oregon. Reasons such as these, in fact, are why we also asked for self-report from participants on their criminal behavior since release. Because our last interview was conducted at 6 months following release from prison, this was the last opportunity we had to assess this via self-report. From past experience, while participants might be committing crimes after release, we knew from past experience that not very many participants in the sample were likely to be arrested during the first 6 months after release, and this would make it difficult to understand if there were differences in arrests between the conditions. Thus, we extended the length of follow-up with official records data as long as possible given that (a) the length of study funding was only 5 years, and (b) we had multiple cohorts that entered the study across the course of the study, so the last cohorts were enrolled during year 4 of the study. These factors led to a decision to collect records data for one year after release from prison.

  1. We have used self-report measures of criminal behavior across four decades of longitudinal research in the center where this study was conducted, the non-profit Oregon Social Learning Center in Eugene, Oregon. We have found good convergence between self-report measures and arrest records in numerous studies. We hypothesize that the self-report data represent a good estimate of illegal behavior, at least for most participants. We have no doubt, however, that for some participants, the data are less accurate. Of course, such a statement is true for any measure and with any sample – neither the reliability nor the validity of measurement are ensured.

  1. The research tradition that this study was centered in – studies of cognitive-behavioral parent training – has not focused on measuring traditional psychological variables. This work arose out of applied behavior analysis and measures of interest are typically behavioral in nature. Further, the questions of interest for this particular study did not focus on impacting psychological variables or on examining psychological variables as moderators or mediators of effects. Some information about background variables that are related to psychological variables is provided in our prior papers (e.g., Kjellstrand et al., 2012).

  1. The majority of the discussion focuses on ideas that we have regarding how to improve programs for incarcerated parents (see Discussion section).